# Measurement of nanoscale three-dimensional diffusion in the interior of living cells by STED-FCS

Luca Lanzanò[1], Lorenzo Scipioni[1,2], Melody Di Bona[1,3], Paolo Bianchini [1,4], Ranieri Bizzarri[1,5], Francesco Cardarelli[6,8], Alberto Diaspro [1,3,4] & Giuseppe Vicidomini [7]

The observation of molecular diffusion at different spatial scales, and in particular below the optical diffraction limit (<200 nm), can reveal details of the subcellular topology and its functional organization. Stimulated-emission depletion microscopy (STED) has been previously combined with fluorescence correlation spectroscopy (FCS) to investigate nanoscale diffusion (STED-FCS). However, stimulated-emission depletion fluorescence correlation spectroscopy has only been used successfully to reveal functional organization in two-dimensional space, such as the plasma membrane, while, an efficient implementation for measurements in three-dimensional space, such as the cellular interior, is still lacking. Here we integrate the STED-FCS method with two analytical approaches, the recent separation of photons by lifetime tuning and the fluorescence lifetime correlation spectroscopy, to simultaneously probe diffusion in three dimensions at different sub-diffraction scales. We demonstrate that this method efficiently provides measurement of the diffusion of EGFP at spatial scales tunable from the diffraction size down to ~80 nm in the cytoplasm of living cells.

[1] Nanoscopy, Nanophysics, Istituto Italiano di Tecnologia, via Morego 30, Genoa 16163, Italy. [2] Department of Computer Science, Bioengineering, Robotics and Systems Engineering, University of Genoa, Genoa 16145, Italy. [3] Department of Physics, University of Genoa, via Dodecaneso 33, Genoa 16146, Italy. [4] Nikon Imaging Center, Istituto Italiano di Tecnologia, via Morego 30, Genoa 16163, Italy. [5] NEST, Scuola Normale Superiore and Istituto Nanoscienze, CNR (NANO-CNR) piazza San Silvestro 12, Pisa 56127, Italy. [6] Center for Nanotechnology Innovation @NEST, Istituto Italiano di Tecnologia, piazza San Silvestro 12, Pisa 56127, Italy. [7] Molecular Microscopy and Spectroscopy, Nanophysics, Istituto Italiano di Tecnologia, via Morego 30, Genoa 16163, Italy. [8] Present address: NEST, Scuola Normale Superiore and Istituto Nanoscienze-CNR, Piazza San Silvestro 12, Pisa 56127, Italy. Correspondence and requests for materials should be addressed to L.L. (email: Luca.Lanzano@iit.it) or to A.D. (email: Alberto.Diaspro@iit.it) or to G.V. (email: Giuseppe.Vicidomini@iit.it)

Fluorescence correlation spectroscopy (FCS) is a powerful statistical tool for characterizing diffusion and kinetics of fluorescently labeled molecules in solution[1, 2]. Its popularity and range of applications grew substantially with the advent of confocal and two-photon excitation microscopy, which allow the observation of small femtoliter volumes at the surface or within the interior of cells. In truth, confocal microscopy opened the possibility of applying the FCS technique to measure biological molecular mobility and how this mobility relates to the cell physiology[3, 4].

More recently, FCS has been combined with stimulated-emission depletion (STED) microscopy[5] (STED-FCS) to study diffusion of molecules at a spatial scale well below the limit imposed by the diffraction of light[6–8]. Diffraction of light limits the observation volume of a confocal microscope to around 200 nm in the $x$–y plane and around 1 µm along the optical axis; STED microscopy uniquely can reduce this volume to tens of nanometers size along all directions. Additionally, since in STED microscopy the size of the observation volume can be tuned easily and continuously either by changing the intensity of the so-called STED beam (conventional STED-FCS)[7] or by analyzing fluorescence dynamics (gated STED-FCS)[9–12], STED-FCS allows the probing of molecular mobility at a decreasing scale, from diffraction size to tens of nanometers. The plot of the transit time $t_D$ of a molecule through the observation volume measured by FCS as a function of the observation area (the so-called FCS diffusion law), has been shown to be fundamental for discriminating between different types of motion, as for instance free diffusion vs. diffusion confined by microdomains[13].

In the cell membrane, STED-FCS has enabled the direct observation of millisecond anomalous lipid and protein dynamics on spatial scales down to ~30 nm[7, 9, 14]. Furthermore, thanks to the ability to record the full diffusion law in a single measurement, gated STED-FCS has revealed the spatial heterogeneity of the cell membrane[10]. The scenario is more complex when STED-FCS needs to investigate mobility of molecules that are not moving in a two-dimensional (2D) space, such as the cellular membrane, but within a three-dimensional (3D) space, such has the cellular interior. As a matter of fact, even if STED microscopy is fully compatible with 3D samples, as it has been largely demonstrated for imaging[15], direct observations of nanoscale diffusion in 3D environments by STED-FCS have been quite limited[6, 8, 16–18], and mostly do not deal with the analysis of molecular diffusion within the cell interior.

The main problem encountered when performing STED-FCS in 3D is a significant increase in unspecific background signal that damps the correlation amplitude and precludes accurate FCS measurements[6, 8]. The expected outcome of a STED-FCS experiment is that both the average transit time $t_D$ of the molecules through the observation volume and the average number $N$ of molecules in the observation volume decrease with an increasing STED beam intensity. However, for fluorophores diffusing freely in 3D in solution, only the expected decrease of $t_D$ is observed, whereas $N$ does not decrease accordingly[6, 8]. This issue has been ascribed to a reduced signal-to-background ratio caused by non-depleted, low-brightness fluorescence signal from out-of-focus volume shells[6]. For a proper determination of the particle number it has been shown that by using information from the fluorescence intensity distribution analysis (FIDA) it is possible to determine the low brightness fraction and correct the FCS data. However this optimized analysis introduces a rather complex global fitting of the autocorrelation function (ACF) and of the photon histogram data and, as far as we know, it has been validated only on fluorophores diffusing freely in aqueous solution and not for investigating molecular mobility in the cell

interior[6]. An alternative strategy is trying to determine this background directly and subtracting it from the total signal[18].

Recently we have introduced a method for super-resolution imaging, called separation of photons by lifetime tuning (SPLIT), based on the explicit separation of the position-dependent fluorophore dynamics generated in a continuous-wave (CW)-STED microscope[19]. A distinctive feature of the SPLIT method, with respect to other time-resolved STED methods[9, 20], is the isolation of uncorrelated background[19]. This feature makes SPLIT particularly attractive for STED-FCS in 3D. In CW-STED-FCS the two most common sources of temporally uncorrelated background are (i) the detector afterpulse and (ii) the anti-Stokes fluorescence emission induced directly by the STED beam[21]. Detector afterpulse can be removed by cross-correlating the signal from two detectors[22], whereas STED-induced background is more difficult to deal with. For instance, in gated STED-FCS the temporal gate is not sufficient to eliminate the uncorrelated background from the signal[21].

Here, we adapt the SPLIT method to work in synergy with fluorescence lifetime correlation spectroscopy (FLCS)[23] to perform background-unbiased STED-FCS in 3D. The SPLIT-FLCS combination allows probing, from a single-FCS experiment, the diffusion from uncorrelated background-free and continuously decreasing in size nanoscale observation volumes in a 3D environment. As a matter of fact, the ability of SPLIT-FLCS to generate the diffusion law at a single point and in a relatively short time can potentially reveal spatial and temporal heterogeneity of molecular mobility in the cell. We find that, for a given STED power, the maximum achievable spatial resolution of a SPLIT-FLCS measurement is limited by the signal-to-noise ratio. Furthermore, in contrast to the gated STED-FCS method the proposed SPLIT-FLCS method is also able to automatically estimate the particle number at each spatial scale without using FCS-FIDA analysis or separate background determination. As an application of SPLIT-FLCS, we measure the diffusion of enhanced green fluorescent protein (EGFP) at spatial scales tunable from the diffraction size down to ~80 nm range in the 3D interior of living cells. We show the spatial heterogeneity in diffusion of EGFP and tubulin-EGFP within the cytoplasm of living cells.

## Results

**The SPLIT-FLCS method.** The method is implemented on a STED microscopy configuration that uses pulsed excitation and CW-STED beam (Fig. 1a). The fluorescence decay, typically occurring on the nanosecond time scale, is observed inside a temporal window defined by the period of excitation $T$ ($T = 12.5$ ns). The doughnut-shaped, CW-STED beam affects the fluorescence decay rate of each fluorophore in a measure proportional to the STED beam intensity. As a result, the fluorescence lifetime $\tau$ is a function of the position of the fluorophore within the observation volume, being maximum at the center, and decreasing towards the periphery (Fig. 1a). We describe the spatial variations of lifetime using the simple model introduced for the SPLIT method:[19]

$$\frac{1}{\tau} = \frac{1}{\tau_0}\left(1 + k_S \frac{r^2}{w_0^2}\right) \qquad (1)$$

where $\tau_0$ is the unperturbed excited-state lifetime of the fluorophore, $r$ is the radial distance from the optical axis, $w_0$ is the waist of the Gaussian observation volume, or point spread function (PSF), of the confocal microscope, and $k_S$ is a parameter that quantifies the relative variation of decay rate values within the observation volume. The parameter $k_S = I_{STED}(w_0)/I_{SAT}$ is the ratio between the value of STED intensity at radial position $r = w_0$ and the saturation value $I_{SAT}$ for which the probability of decay

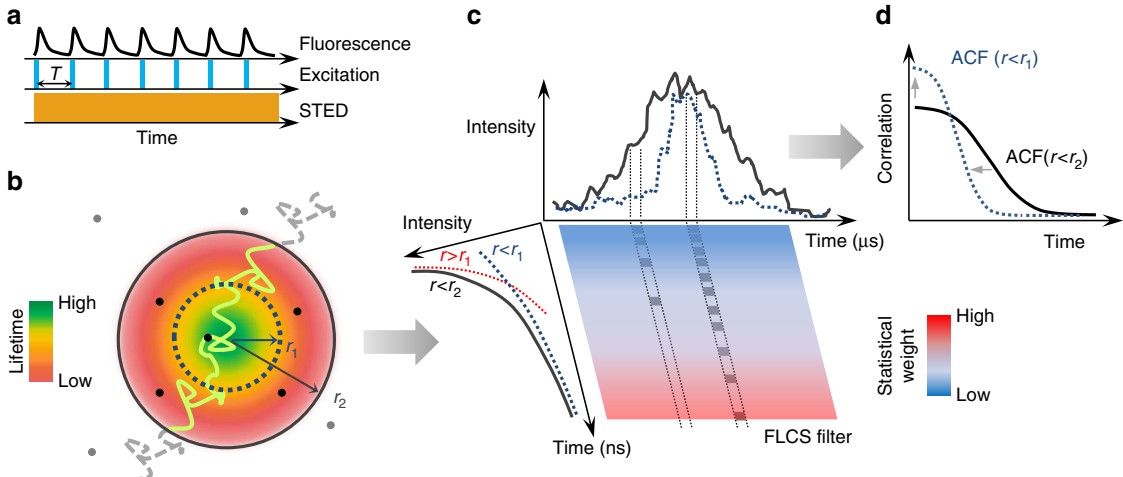

**Fig. 1** Schematic principle of the SPLIT-FLCS method. **a** A pulsed laser excitation source is coupled with a CW-STED beam to exploit the variations in the fluorescence dynamics observed in the nanosecond time scale ($t < T$). **b** The spatial variations of fluorescence lifetime within the effective observation volume of the microscope can be modeled as a gradient in the radial direction, with the lifetime value being the highest at the center and decreasing towards the periphery. A molecule transiting within the observation volume will emit photons with different fluorescence lifetime according to the radial position. **c** The temporal fingerprints associated to different spatial positions can be used to weight the photons based on their time of arrival in the nanosecond time scale. For instance, the decay associated to the region defined by radial position $r < r_1$ can be used to generate a statistical filter that, by over-weighting the late-arriving photons, sort out only the intensity emitted from a smaller effective observation volume. **d** Correlation of the weighted photons on the time scale relevant for molecular diffusion ($t > 1\,\mu s >> T$) results in a filtered ACF associated to fluctuations on a smaller effective observation volume, namely an ACF which is narrower and has larger amplitude

due to stimulated and spontaneous emission are equal[19]. The precise value of $k_S$ depends on the optical configuration, i.e., the intensity distributions, and on the properties of the sample[19]. In order to estimate the value of $k_S$ for a given optical configuration and for a given sample, we introduced in ref. [19] an analytical model for the fluorescence intensity decay corresponding to the lifetime distribution described by Eq. (1). According to this model the decay of the fluorescence intensity $J_{STED}(t)$ in presence of the STED beam (hereafter referred to as STED decay) can be described by the following formula (Methods section):

$$J_{STED}(t) = I_0 e^{-\frac{t}{\tau_0}} \frac{1}{1 + \frac{k_S}{2}\frac{t}{\tau_0}} + I_0 b \qquad (2)$$

where $b$ represents the fraction of uncorrelated background and $I_0$ the amplitude of the decay. The value of $k_S$ and $b$ for a given STED power and for a given fluorophore can be extracted by measuring the experimental STED decay and by fitting it to Eq. (2). According to the principle of SPLIT-based super-resolution technique, a smaller observation volume can be obtained by selecting, from the fluorescence signal, only those photons characterized by a specific temporal fingerprint. For instance, the region corresponding to a radial distance $r < r_1$, has a different temporal fingerprint than the region corresponding to a radial distance $r > r_1$. A fluorophore diffusing across the observation volume will emit photons with different temporal fingerprints when transiting across the two regions (Fig. 1b).

FLCS is a method that uses time-resolved detection for separating the FCS contributions associated with different fluorescence lifetime components[23, 24]. FLCS is based on the application of statistical filters or weighting functions to the detected photon counts before calculation of the ACF (Fig. 1c). A filtered ACF associated with the slowest component ($r < r_1$), and thus to a smaller effective observation volume, can be obtained by properly weighting the photons based on their arrival time on the nanosecond temporal scale (Fig. 1c and d). The requirement of FLCS is that a decay pattern or temporal

fingerprint is specified for each component. Here, in order to specify the decay patterns, we set an arbitrary value of the parameter $r_1$ (expressed in units of $w_0$) and calculate the decay patterns associated to the inner ($r < r_1$) and outer ($r > r_1$) spatial components, according to the lifetime distribution described by Eq. (1) (Methods section). By decreasing the value of the parameter $r_1$ we are able to probe FCS on smaller observation volumes.

The uncorrelated background is also included in the FLCS analysis as an additional decay pattern (Methods section). In each experiment, the relative fraction of uncorrelated background with respect to the decaying fluorescence signal is quantified by the parameter $b$ in Eq. (2), which can be determined experimentally. This is a notable difference between gated STED-FCS and SPLIT-FLCS. In gated STED-FCS the observation volume is tuned by changing a temporal detection window that, alone, is not sufficient to remove the uncorrelated background from the signal[21]. In the SPLIT-FLCS method, we are able to isolate the uncorrelated background from the total signal by considering it as an additional fluorescence intensity "decay".

**SPLIT-FLCS of EGFP in aqueous solution.** We demonstrate now the ability of our SPLIT-FCS method to measure the mobility of fluorescent molecules dissolved in an aqueous environment. In particular, we use a solution of free diffusing EGFP, and we show how this measurement is necessary for calibrating our method for successive measurements, for example in a different environment or in the cytoplasm of living cells.

The characterization of the spatial variations of lifetime, induced at a given STED beam intensity level, is obtained from the analysis of the STED decay shown in Fig. 2a. As expected, the STED decays are non-exponential and the values of $k_S$ extracted from the fit to Eq. (2) are proportional to the STED beam intensity (Fig. 2b). Higher values of $k_S$ indicate stronger variations of lifetime across the observation volume. For instance, a value of $k_S = 2$ means that the lifetime $\tau$ of EGFP molecules located at a

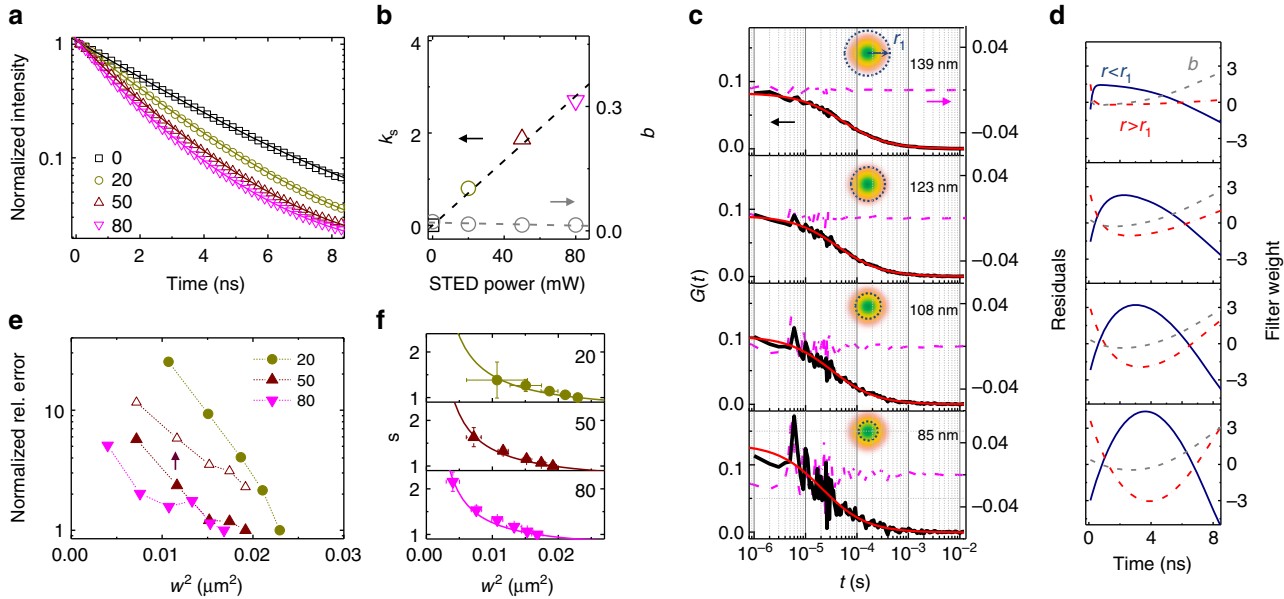

**Fig. 2** SPLIT-FLCS of EGFP in solution. **a** Experimental STED decays of EGFP in PBS for different levels of STED intensity. Numbers indicate STED beam power in mW (measured at the back aperture of the objective lens). The solid lines represent a fit of the data to the Eq. (2). Value of $\tau$ recovered from the confocal measurement is $\tau_{EGFP} = 2.6$ ns. **b** Values of the parameters $k_s$ and $b$ recovered from the fit shown in **a**. The value of $k_s$ vs. STED power indicates an increase of the spatial variations of lifetime within the PSF. **c** Filtered ACFs of EGFP in PBS obtained at a STED power of 50 mW for different values of the parameter $r_1$ ($r_1 = 11w_0$, $2w_0$, $w_0$, $0.5w_0$, respectively, from top to bottom). Each of the ACFs is obtained from a FLCS-based separation into three components **d** and is associated to the calculated decay corresponding to the radial region $r < r_1$. Solid red lines are a fit of the data to Eq. (15) and the numbers indicate the recovered lateral size $w$ of the effective observation volume. Residuals are shown on the right axis (*purple dashed line*). **d** Filter weighting functions corresponding to the temporal signatures of the fluorophores located at distance $r < r_1$ (*solid blue line*), the fluorophores located at distance $r > r_1$ (*red dash-dot line*), and the fraction $b$ of uncorrelated background (*gray dashed line*), respectively. **e** Normalized relative error associated to the values of $w^2$ recovered from SPLIT-FLCS analysis of EGFP at different STED intensity levels (STED beam power indicated in mW). The increase in relative error when the measurement time is reduced to $t_{acq} = 20$ s is reported only for the STED power of 50 mW (*open triangles*). **f** Calibration of the effective volume shape factor $s$ at different STED intensity levels (mean ± s.d.). The solid lines are a fit of the data to Eq. (22). Values of $\xi_z$ recovered from the fit are $\xi_z = 0.63 \pm 0.04$, $0.60 \pm 0.03$, $0.62 \pm 0.02$ for STED powers 20, 50, 80 mW, respectively

radial distance $r = w_0/2$ is $(1 + k_S r^2/w_0^2) = 1.5$ times shorter than the lifetime $\tau_0$ of those located at the center of the observation volume. Provided the values of $k_S$ and $\tau_0$ are determined from calibration, Eq. (1) describes the distribution of decay rates as a function of the radial position in the observation volume. The fraction $b$ of uncorrelated background (detector afterpulse and/or STED-induced fluorescence emission) is also determined from the STED decay and Eq. (2) (Fig. 2a and b).

We then calculate the temporal signature of the fluorophores located at distance $r < r_1$, and, for different values of $r_1$, we generate the corresponding filtered ACFs (Fig. 2c). Note that, for each value of $r_1$, the FLCS analysis is performed by defining three temporal signatures: (i) the expected decay $J_1$ from all the fluorophores located at distance $r < r_1$, (ii) the expected decay $J_2$ from all the fluorophores located at distance $r > r_1$, (iii) the fraction $b$ of uncorrelated background. These three components define three corresponding filter functions (Fig. 2d) that, for each value of $r_1$, generate three filtered ACFs (Supplementary Fig. 1). We are only interested in the filtered ACF associated with the first of these three temporal signatures (Fig. 2c). As we decrease the value of $r_1$, the filtered ACFs, obtained from the same measurement, decay faster and are characterized qualitatively by larger amplitude and lower signal-to-noise ratio (Fig. 2c, Supplementary Fig. 2).

Notably, while in SPLIT-based super-resolution imaging the fluorescence photons sorting (between the sub-diffraction volume, the external shell, and the background) is obtained through the phasor analysis, in the SPLIT-FLCS method there is

not a real photons sorting, but each photon contributes to the different ACF curve according to the weight established by the filters of the FLCS approach.

By assuming a diffusion coefficient $D = 90 \; \mu m^2 \; s^{-1}$ for free EGFP in solution, we can calibrate the value of waist $w$ associated to the effective observation volumes. For relatively large values of $r_1$, the filter operates only to remove the uncorrelated background and we obtain a value $w = w_{STED}$, which is smaller than the confocal value ($w_0 = 160$ nm) and depends on the level of the STED power (Fig. 2c and d, Supplementary Fig. 2). For decreasing values of $r_1$, as we start to separate modulated fluorescence photons based on their dynamics, we are able to get increasingly smaller values of $w$ ($w < w_{STED}$) from the very same set of data (Fig. 2c, Supplementary Fig. 2). We observe that the action of the filter for decreasing values of $r_1$ results in an amplification of the noise of the ACF (Fig. 2c, Supplementary Fig. 2), quantified through the relative error of $w^2$ (Fig. 2e). As expected, noise amplification is larger at lower STED powers, where the STED-induced spatial variations of lifetime are less pronounced. In order to quantify which is the smallest value of $w$ that can be obtained at a given STED power, we need to specify the required level of SNR. For instance, in the specific conditions of the experiment shown in Fig. 2c, at a STED power of 50 mW we get a relative uncertainty of ~3% in the determination of $w^2_{STED}$. If we require a maximum relative uncertainty of ~20% in the value of $w^2$ then the smallest value that we can get is $w$~80 nm (Fig. 2e). Finally, another factor to take into account is the acquisition time: for instance, for the measurement at a STED power of 50 mW, reducing the acquisition time $t_{acq}$ from 100 to

20 s causes a reduction in signal-to-noise ratio that will limit the smallest value that we can get to $w \sim 105$ nm (Fig. 2e).

Since at the used STED beam wavelength (577 nm) direct excitation of EGFP is negligible[25], the uncorrelated background in these experiments is relatively low ($b < 0.03$) also for increasing STED beam power. In order to check that the removal of background works efficiently in more critical experiments, namely when the total amount of uncorrelated photon counts is comparable with the total amount of photon counts from the decaying fluorescence signal, we also performed tests using another fluorophore (Supplementary Fig. 3).

Next we checked if the decrease of the lateral size $w$ of the effective observation volume corresponded to a decrease in the number of molecules $N$ calculated from the amplitude $G(0)$ of the ACFs (Methods section). A non-linear scaling emerges by plotting the number of molecules $N$ as a function of the lateral size $w^2$ (Supplementary Fig. 4a). In order to correct the number of molecules $N$, we introduce now a geometrical correction factor based on the following considerations. So far, our analysis has been based on a description of fluorescence lifetime variations (Eq. 1) obtained in the framework of a simple 2D approximation that reflects the average pattern of the STED beam intensity distribution at axial positions near the focal plane (Supplementary Fig. 5a). At axial positions far from the focal plane, the reduced intensity of the STED beam is expected to induce little or no variations of lifetime (Supplementary Fig. 5a). Notably, the intensity contribution from these out-of-focus planes is characterized by a much lower brightness and does not prevent the capability of generating smaller effective observation volumes from the point of view of fluorescence fluctuations (i.e., smaller values of $w$). On the other hand this low-brightness out-of-focus contribution may affect the shape of the 3D observation volume from the point of view of the measured value of particle number. Indeed, say $N = \rho V$ is the average number of molecules present at a concentration $\rho$ in the volume $V$. Since the action of the STED beam is less efficient in the out-of-focus planes (Supplementary Fig. 5a), the reduction in volume $V/V_{\text{STED}}$ is less pronounced compared to the reduction of the lateral size $w^2/w^2_{\text{STED}}$ (Supplementary Fig. 5b and c). In other words, our cylindrical description of the observation volume partially fails and a dim fluorescent signal stemming from out-of-focus planes acts as a background reduction in the amplitude of the ACF. Here, we define a correction factor $s > 1$

$$\frac{V}{V_{\text{STED}}} = s \frac{w^2}{w^2_{\text{STED}}} \qquad (3)$$

that, assuming uniform concentration in the solution, can be calibrated directly from the same experimental data (Fig. 2f, Eq. 20). After the application of the correction factor $s$ to the observation volume, the real particle number $N^{\#}$ can be easily estimated (Supplementary Fig. 4b). Importantly, we derive an easy model describing the behavior of the correction factor $s$ as a function of the lateral size $w$, based on a realistic geometrical shape of the effective observation volume (Supplementary Note 1 and Supplementary Fig. 5b and c). The experimental values of $s$ are in perfect agreement with Eq. (22), which was derived within this model (Fig. 2f). Compared to the more rigorous FCS-FIDA analysis[6], here we correct the value of particle number by simply calibrating the asymmetry between in- and out-of-focus regions of the observation volume (Supplementary Fig. 5). It would be interesting to check if a similar correction works also on STED configurations able to generate a more isotropic lifetime pattern in 3D[8, 26].

The values of $w$ and $s$ obtained from the measurement of EGFP in aqueous solution can be used as calibration values to measure the multiscale diffusion properties of EGFP in another medium, for instance a more viscous solution (Supplementary Fig. 6), but more importantly to measure the mobility of molecules in the interior of living cells.

**SPLIT-FLCS in the cytoplasm of live cells**. As an application of the 3D SPLIT-FCS method, we measured the nanoscale diffusion of EGFP within the 3D cellular interior by performing SPLIT-FLCS on single-points in the cytoplasm of HeLa cells (Fig. 3, Supplementary Fig. 7). An example of this type of measurement, performed at a STED power of 50 mW, is reported in Fig. 3a. The filtered ACFs, obtained by decreasing the value of $r_1$, are characterized qualitatively by the same behavior observed for EGFP in solution, namely smaller width, larger amplitude, and lower signal-to-noise ratio. Using the values of $w$ and $s$ obtained from the calibration with EGFP in solution one can retrieve the transit time $t_D$ and the apparent diffusion coefficient $D$ (Fig. 3b) at different spatial scales (different area $w^2$). By plotting the corrected particle number $N^{\#}$ as a function of the observation area $w^2$ (Fig. 3c) we observe linear trends of very different slopes, due to the variations in the level of EGFP expression from cell to cell, with intercepts close to zero. A similar analysis has also been performed on HeLa cells at a lower STED beam power (20 mW, Supplementary Fig. 7) and on a different cell line (Supplementary Fig. 8). On average, we observed an almost linear dependence of $t_D$ as a function of the observation area $w^2$ with a coefficient $D = 25 \ \mu m^2 \ s^{-1}$, across multiple spatial scales ranging from $\sim 140$ to $\sim 80$ nm (Fig. 3d). The recovered value of $D$ is in keeping with previous reports from confocal FCS in the cytoplasm of HeLa cells[27, 28]. A similar behavior and a similar value for the diffusion coefficient ($D = 26$ $\mu m^2 \ s^{-1}$) are obtained from the measurements performed on HeLa cells at lower STED power (Supplementary Fig. 7). A similar behavior but a different value of diffusion coefficient ($D = 38 \ \mu m^2 \ s^{-1}$) are obtained from the measurements performed on CHO cells (Supplementary Fig. 8). This value is in keeping with recently reported measurements of EGFP mobility in CHO cytoplasm obtained by the analysis of fast spatio-temporal correlation functions at tunable temporal scales[29]. Although this latter approach was able to probe diffusion of EGFP on an even shorter spatial scale compared to what attained here (i.e., down to $\sim 25$ nm)[29], the diffusion parameters were averaged over areas of the sample with a size of at least a few microns. Compared to that, the present approach extracts diffusion parameters from a single point, opening the opportunity of detecting heterogeneities between different intracellular locations.

The spatial heterogeneity is visible for instance in the variability between the single-point diffusion laws reported in Fig. 3d and e, Supplementary Figs. 7b and 8b. In Fig. 3d we report, for the sake of clarity, only three representative measurements, whose linear trends intersect the time axis at positive, null, and negative values, thus highlighting the heterogeneity in the diffusion mode of EGFP within the cytoplasm. Fig. 3f and g shows the results of three fast measurements ($t_{\text{acq}} = 20$ s) performed consecutively on the very same point, showing the persistence of a specific diffusion mode in that particular location. Heterogeneity emerges also in the relatively large error bar in the corresponding average diffusion laws (Fig. 3d, Supplementary Fig. 7d and 8d) when compared with the average diffusion laws measured in a viscous EGFP solution (Supplementary Fig. 5d).

Next we performed the same type of measurements on another protein, tubulin-EGFP, in the cytoplasm of HeLa cells (Fig. 4, Supplementary Fig. 9). Tubulin is the building block of microtubules and is composed of a heterodimer of two closely related forms ($\alpha$ and $\beta$ tubulin). Characterizing diffusion of

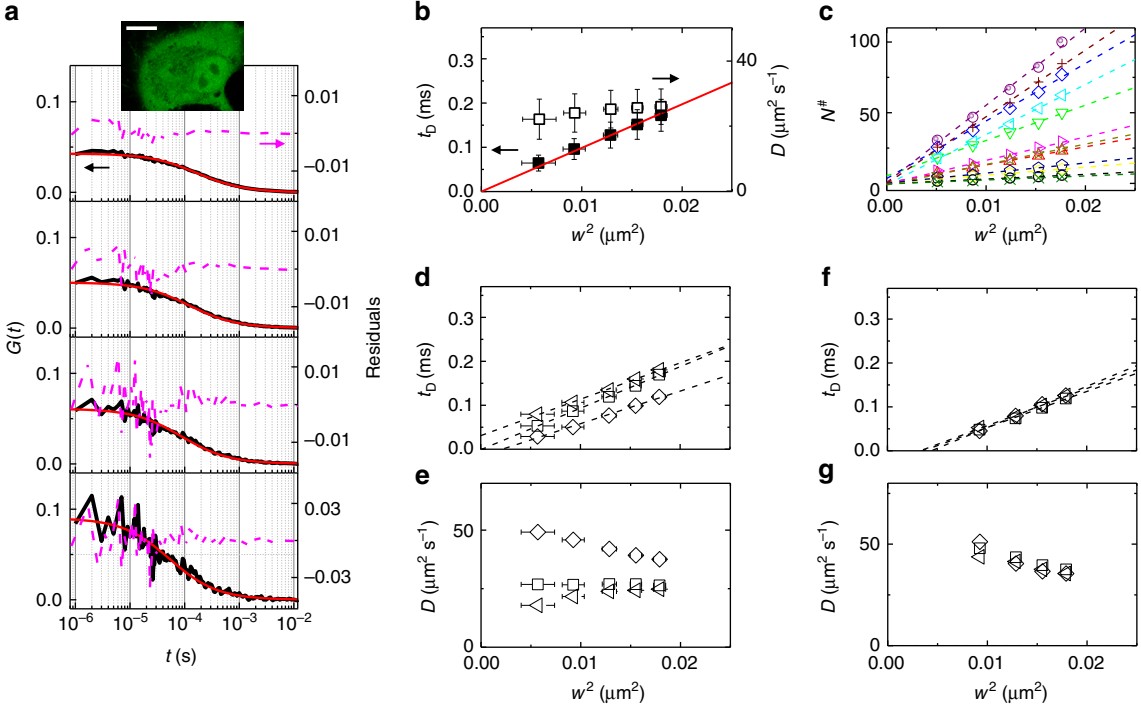

**Fig. 3** SPLIT-FLCS of EGFP in the cytoplasm of live cells. **a** Representative filtered ACFs of EGFP in the cytoplasm of a HeLa cell at a STED power of 50 mW for different values of the parameter $r_1$. Solid lines are a fit of the data to Eq. (15). Residuals are shown on the right axis. The image of a cell is shown in the inset (scale bar 10 μm). **b** Transit time $t_D$ (*solid squares*) and apparent diffusion coefficient $D$ (*open squares*) for different spatial scales, obtained as the average of single-point measurements in the cytoplasm of $n = 12$ different HeLa cells (mean ± s.d.). The solid line is a fit to Eq. (14) ($D = 25 \, \mu m^2 \, s^{-1}$). **c** Corrected value of the number of molecules $N^\#$ vs. $w^2$, for all the measured cells. The dashed lines are linear fits to the data. The slope varies from cell to cell according to differences in EGFP expression level. **d** Transit time $t_D$ and **e** apparent diffusion coefficient $D$ at different spatial scales for three different cells. **f** Transit time $t_D$ and **g** apparent diffusion coefficient $D$ as measured on the very same spot by three short ($t_{acq} = 20 \, s$) consecutive acquisitions

tubulin can be important for the studies of microtubule dynamics[30]. Two representative single-point SPLIT-FLCS measurements of tubulin-EGFP are reported in Fig. 4a. On average ($n = 32$) we observed a scale-dependent (from ~140 to ~100 nm) diffusion that cannot be described as pure Brownian motion (Fig. 4b). The positive intercept, in fact, is a hallmark of the trapping of the tubulin molecules during their motion[13]. This trapping results in an increasing apparent diffusion coefficient $D$ at smaller spatial scales. Interestingly, tubulin shows a more pronounced spatial heterogeneity of its diffusion mode (Fig. 4b and c) as compared to untagged EGFP (Fig. 3b and d). More in detail, we can see that very different diffusion modes can be detected depending on the selected spatial location (Fig. 4c). Using as an arbitrary reference the value of the transit time $t_D$ measured at the smallest spatial scale ($w = 104 \, nm$) we can distinguish a group of faster diffusion modes ($t_D < 0.4 \, ms$) where trapping is negligible, a group of very slow diffusion modes ($t_D > 0.7 \, ms$) where trapping is dominant, and finally a group of intermediate behavior ($0.4 \, ms < t_D < 0.7 \, ms$). It's worth noting that this heterogeneity becomes clear only by probing diffusion at smaller spatial scales. The points characterized by the very slow tubulin diffusion are likely to correspond to intact microtubules where a large fraction of molecules is trapped (and, conversely, a reduced fraction of molecules is mobile). To the best of our knowledge, this is the first time that trapping of tubulin is observed on an arbitrary point in the cytoplasm of live cells by STED-FCS.

## Discussion
The combination of STED microscopy with FCS can offer novel insights into many important cellular processes, since it

allows the direct investigation of molecular mobility at the nanometer spatial scale. This ability has been fully explored to understand the plasma-membrane 2D organization, but applications of the STED-FCS method in a 3D environment, such as the cell interior, have so far encountered technical difficulties.

Here, we have shown that combining STED-FCS with the SPLIT method can solve these difficulties. For the first time, we have shown that it is possible, by a single measurement, to obtain a diffusion law for an arbitrary point within the cell interior. In particular, by using the EGFP as reporter we extracted, from ~100-s long single point FCS measurements inside the cytoplasm, diffusion laws on a spatial scale tunable from the diffraction size down to ~80 nm. Since each diffusion law is a fingerprint of the local nanoscale organization, our approach is able to reveal the spatial heterogeneity of nanoscale organization in a cell interior.

In conventional STED-FCS the reconstruction of the diffusion law may take several minutes, since it requires a single-FCS recording for each STED beam's intensity. Similarly to the gated STED-FCS approaches, in order to obtain different observation volumes from a single measurement, the SPLIT-FLCS method uses the nanosecond fluorescence dynamics information of a time-resolved CW-STED microscopy experiment. In particular, by exploiting the fluorescence photons arrival time information, typical of a time-resolved experiment, it is possible to remove the residual fluorescence stemming from the peripheral shell of the CW-STED microscope's observation volume, and thus tune its effective spatial resolution without increasing the (peak) intensity of the STED beam[9, 19]. Notably, the possibility of working at relatively lower peak intensities represents also an advantage for live-cell conditions, which is fundamental for mobility studies.

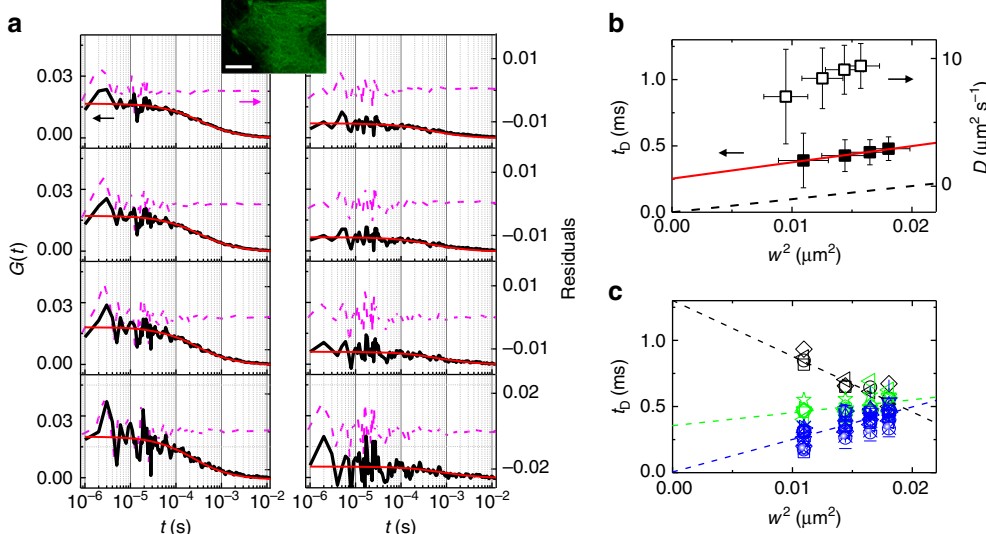

**Fig. 4** SPLIT-FLCS of tubulin-EGFP in the cytoplasm of live cells. **a** Representative filtered ACFs of tubulin-EGFP in the cytoplasm of HeLa cells at a STED power of 50 mW for different values of the parameter $r_1$ ($r_1 = 11w_0$, $4w_0$, $2w_0$, $w_0$, respectively, from *top* to *bottom*). Shown are two experimental measurements taken as representative of fast (*left*) and slow (*right*) diffusion, respectively. Solid lines are a fit of the data to Eq. (15). Residuals are shown on the right axis. The image of a cell is shown in the inset (scale bar 10 μm). **b** Transit time $t_D$ and apparent diffusion coefficient $D$ for different spatial scales, obtained as the average of single-point measurements in the cytoplasm of HeLa cells ($n = 32$, mean ± s.d.). The solid red line is a linear fit to the data. The dashed line represents a Brownian motion with $D = 25$ μm² s⁻¹. **c** Transit time $t_D$ at different spatial scales for all the single-point measurements. The data have been grouped into three categories depending on the value of transit time $t_D$ measured at the smallest spatial scale ($w = 104$ nm): $t_D < 0.4$ ms (blue), $0.4$ ms $< t_D < 0.7$ ms (green), $t_D > 0.7$ ms (black). The *dashed lines* are linear fits of the average of each group

The main novelties of the SPLIT-FLCS method, compared to other gated STED-FCS approaches[9, 10], are (i) a quantitative description of the spatial variations of fluorescence lifetime induced by the CW-STED beam; (ii) the removal of any uncorrelated background from the analysis; (iii) a simple correction model for the particle number in 3D.

In order to fully understand the performances of the method, we need to consider which parameters ultimately determine the smallest achievable spatial scale without increasing the acquisition time. The smallest value of $w$ that we can get depends on the STED beam intensity level and on the SNR of the measurement. The SNR of the measurement, in turn, depends on the brightness of the molecule and on the acquisition time[31–34]. Therefore, for a given acquisition time, a smaller value of $w$ can be obtained by (i) increasing the STED beam power and/or by (ii) increasing the brightness of the molecule. Increasing the STED or excitation beam power is limited by the potential photo-damage induced on the fluorophores on the cells. For instance, in the conditions of our experiments, we observed some photobleaching of EGFP during the acquisition (Supplementary Fig. 10). Photobleaching resulted in a transient concentration gradient between the cytoplasmic and nuclear compartment (Supplementary Fig. 11). Importantly, the photobleaching-induced intensity variations were relatively slow and did not affect the shape of the ACF curves. In view of the potential side effects related to the use of high STED beam illumination powers, it is of primary importance to maximize the brightness of the molecules. In our experimental conditions, we estimate for EGFP a brightness $\varepsilon \sim 25$ kHz per molecule. Any further increase in molecular brightness will help to improve the overall performances of the method, in terms of a smaller effective observation volume, a reduced acquisition time or a lower level of STED beam illumination intensity. Besides obvious instrumental considerations (e.g., well co-aligned STED and excitation beams, optimized detection efficiency), these results suggest that the use of fluorophores that are brighter and more photostable than EGFP might significantly improve the

performance of the method. In this respect, the great potential of silicon rhodamine fluorophores for STED imaging has recently been proved[35]. Their superior brightness and photostability together with their cell permeability and water solubility make these fluorophores perfect candidates for our SPLIT-FLCS approach.

Finally, even if we have limited our characterization to single-point measurements, the method is in principle compatible with any technique extending the FCS analysis to multiple points[4]. In the cell interior, this combination may offer several advantages. For instance, the analysis of fast spatio-temporal correlation functions between multiple observation spots has been used recently to obtain accurate diffusion laws of molecules and to provide fine details about the network of obstacles present in the cytoplasm[29, 36]. These methods would benefit from a significant reduction in the size of the observation volume since they would be able to access even smaller length scales. However, these are intrinsically "average" methods, in the sense that the fine topological details that they provide are extracted from areas of the sample with a size of at least a few microns. Other methods based on scanning-FCS[37–39] have been designed to map diffusion coefficients at different spatial locations and detect spatial heterogeneities. In the plasma membrane, STED-FCS has already been coupled to line or raster scanning to take advantage of self-calibration[11] and to reveal the spatial and temporal heterogeneity of lipid interactions[40, 41]. We believe that, if coupled efficiently with scanning-FCS and proper analysis tools[42], our SPLIT-FLCS method has the potential to explore spatial and temporal heterogeneities in the cell interior, especially in the vicinity of small subcellular structures[43].

## Methods

**SPLIT-FLCS algorithm**. Analysis of the STED decay: The average fluorescence decay associated with each acquisition has been fitted to Eq. (2). This equation represents the integral of single exponential decays distributed according to Eq. (1)

and weighted with a Gaussian PSF:[19]

$$J_{\text{STED}}(t) \propto \int_0^\infty dr^2 e^{-\frac{t}{\tau(r^2)}} e^{-\frac{2r^2}{w_0^2}} \propto e^{-\frac{t}{\tau_0}} \frac{1}{1 + \frac{k_S}{2} \frac{t}{\tau_0}} \qquad (4)$$

Equation (2) allows the retrieving of the values of $k_S$, $\tau_0$ and $b$. For confocal measurements (i.e., measurements acquired without the STED beam) the value of $k_S$ has been fixed to 0.

In the case of EGFP, the parameters $k_S$, $\tau_0$, and $b$ used for the analysis in solutions and in cells have been determined as follows. The lifetime value $\tau_{\text{EGFP}}$ for EGFP in PBS has been determined from confocal measurements. Then, the value of $k_S$ for EGFP in PBS at a given STED power has been determined from the STED measurements, by fixing $\tau_0$ to the value $\tau_{\text{EGFP}}$. For the STED measurements of EGFP in cells, the value of $k_S$ was fixed to the value determined from EGFP in solution at the same STED power: this is equivalent to assuming that the STED saturation intensity of the fluorophore $I_{\text{SAT}}$ is the same in solution and in cells. The value of $\tau_0$ was determined independently in each measurement to take into account potential variations associated with the intracellular environment and/or photobleaching[44, 45]. Finally, the value of $b$ was evaluated independently for each measurement.

Calculation of the decay patterns: The temporal decay patterns are calculated analytically starting from the parameters $k_S$ and $\tau_0$. For a given value of $r_1$, we calculate three temporal decay patterns, $J_1$, $J_2$, and $J_3$. The decays $J_1$ and $J_2$, associated to the fluorophores located at distance $r < r_1$ and $r > r_1$, respectively, are calculated as:

$$J_1(t) \propto \int_0^{r_1} dr^2 e^{-\frac{t}{\tau(r^2)}} e^{-\frac{2r^2}{w^2}} \propto e^{-\frac{t}{\tau_0}} \frac{1}{1 + \frac{k_S t}{2\tau_0}} \left( 1 - e^{-\left(1 + \frac{k_S t}{2\tau_0}\right) 2r_1^2/w^2} \right) \qquad (5a)$$

$$J_2(t) \propto \int_{r_1}^\infty dr^2 e^{-\frac{t}{\tau(r^2)}} e^{-\frac{2r^2}{w^2}} \propto e^{-\frac{t}{\tau_0}} \frac{1}{1 + \frac{k_S t}{2\tau_0}} e^{-\left(1 + \frac{k_S t}{2\tau_0}\right) 2r_1^2/w^2} \qquad (5b)$$

The corresponding normalized temporal decay patterns are calculated as:

$$\hat{J}_1(t) = J_1(t) / \int_0^T J_1(t) dt \qquad (6a)$$

$$\hat{J}_2(t) = J_2(t) / \int_0^T J_2(t) dt \qquad (6b)$$

where $T$ is the inverse of the excitation pulse repetition rate. The normalized temporal signature of the uncorrelated background, used as a third component, is defined as:

$$\hat{J}_3(t) = \frac{1}{T} \qquad (7)$$

Calculation of the filter weight functions: The filter functions $F_{ik}$ used to weight photons according to their time of arrival have been calculated following the algorithm described by Kapusta et al.[23]. First, a matrix $\mathbf{M}$ containing the normalized decay patterns is calculated as:

$$M_{ik} = \hat{J}_i((k-1)\Delta t) \qquad (8)$$

where $k = 1,\ldots, N_{\text{bin}}$ indicates the binning of the time range $[0,T]$ into $N_{\text{bin}}$ intervals and $\Delta t = T/N_{\text{bin}}$ is the bin width of the photon arrival time histogram ($\Delta t = 0.1$ ns). Then the diagonal elements of a diagonal matrix $\mathbf{D}$ are calculated as:

$$D_{kk} = J_{\text{STED}}^{-1}((k-1)\Delta t) \qquad (9)$$

where $J_{\text{STED}}(t)$ is the experimental STED decay described by Eq. (2) and the experimental parameters $k_S$, $\tau_0$, and $b$. Finally, a filter function matrix is calculated according to the formula:

$$\mathbf{F} = \left[ \mathbf{M} \cdot \mathbf{D} \cdot \mathbf{M}^T \right]^{-1} \cdot \mathbf{M}^T \cdot \mathbf{D} \qquad (10)$$

The filter functions are defined as:

$$F_i((k-1)\Delta t) = F_{ik} \qquad (11)$$

Calculation of the filtered ACF: The filtered ACF corresponding to the $i$th component is calculated as:

$$G_i(t) = \frac{\left\langle \sum_k F_{ik} I_k(t') \sum_k F_{ik} I_k(t' + t) \right\rangle}{\left\langle \sum_k F_{ik} I_k(t') \right\rangle^2} - 1 \qquad (12)$$

where $I_k(t')$ denotes the intensity value detected at time $t'$ with photon arrival time $(k-1)\Delta t$, and the brackets indicate averaging over all values of time $t'$ (measured on a time scale above a microsecond). The ACFs have been calculated for values of the correlation time $t$ between 1 μs and 0.5 s.

Model for ACF analysis: The filtered ACFs have been fitted to a 3D Gaussian diffusion model:

$$G(t) = G(0) \left( 1 + \frac{t}{t_D} \right)^{-1} \left( 1 + \frac{t}{k_z t_D} \right)^{-1/2} \qquad (13)$$

where $G(0)$ is the amplitude of the ACF, $k_z = w_z/w$ is the ratio between the size of the effective volume along $z$ and $x$–$y$, respectively, and $t_D$ is the characteristic transit time:

$$t_D = \frac{w^2}{4D} \qquad (14)$$

In our setup, $k_z \gg 1$, so that we can use the approximate formula:

$$G(t) \approx G(0) \left( 1 + \frac{t}{t_D} \right)^{-1} \qquad (15)$$

Equation 14 can be used to calibrate $w$ or to determine $D$. The amplitude $G(0)$ is inversely proportional to the number of particles $N$ in the observation volume $V$:

$$G(0) = \frac{\gamma}{N} \qquad (16)$$

where $\gamma$ is the PSF-model dependent gamma factor[46] ($\gamma = 0.35$ for a 3D Gaussian PSF). Equation 16 is used to calculate the non-corrected number of particles $N$.

Model for particle number correction: Since we implemented our STED microscope with a conventional doughnut-shaped STED beam, the STED beam intensity distribution has the same pattern at any axial position along the whole axial extension of the confocal observation volume. Also as a consequence also the lifetime distribution at any axial position is identical and increasing the intensity of the STED beam reduces only the lateral size $w$ of the effective observation volume, leaving the axial extension $w_z$ unchanged. Under these assumptions, the effective observation volume $V$ can be expressed as:

$$V = k_{\text{vol}} w^2 w_z \qquad (17)$$

where $k_{\text{vol}}$ is a constant that depends on the shape of the effective PSF. Then we can rewrite:

$$G(0) = \frac{\gamma}{\rho V} = \frac{\gamma}{\rho k_{\text{vol}} w^2 w_z} \qquad (18)$$

At a given STED power, we consider the amplitude of the filtered ACF obtained for $r_1 \gg w_0$ (STED ACF):

$$G_{\text{STED}}(0) = \frac{\gamma}{\rho V_{\text{STED}}} = \frac{\gamma}{\rho k_{\text{vol}}^{\text{STED}} w_{\text{STED}}^2 w_z} \qquad (19)$$

The shape factor $s$, defined in Eq. (3), can be calibrated using a solution of the fluorophore at uniform concentration, using the formula:

$$s = \frac{k_{\text{vol}}}{k_{\text{vol}}^{\text{STED}}} = \frac{G_{\text{STED}}(0)}{G(0)} \frac{w_{\text{STED}}^2}{w^2} \qquad (20)$$

The calibrated shape factor $s$ is used to calculate the corrected number of particles $N^{\#}$:

$$N^{\#} = \frac{N}{s} \qquad (21)$$

If we assume that the STED is effective only in a region $\Delta_z < w_z$ near the focal plane, the shape factor can be approximated as (see Supplementary Fig. 5 and Supplementary Note 1):

$$s \approx \frac{w_{STED}^2}{w^2}(1 - \xi_z) + \xi_z \qquad (22)$$

where $\xi_z = \Delta_z/w_z$.

All the fitting procedures were performed in OriginPro (OriginLab) using an unweighted least squares procedure.

Experiments: All the experiments were performed on a home-built CW-STED microscope[47]. The excitation beam was provided by a supercontinuum source and the STED beam was provided by a CW Optically Pumped Semiconductor Laser emitting at 577 nm (Genesis, Coherent). We generated the supercontinuum source by pumping a photon-crystal-fiber (femtoWHITE-800, NKT Photonics) with a femtosecond mode-locked Ti:Sapphire laser of 150 fs pulse width, 80 MHz repetition rate (Chameleon, Vision II, Coherent). The STED beam passed through a polymeric mask imprinting $0–2\pi$ helical phase-ramps (VPP-A1, RPC Photonics) in order to obtain a doughnut-shaped diffraction pattern at the focus. The STED and the excitation beams were collinearly aligned using two dichroic mirrors (zt-488-RDC and z-560-sprdc, AHF analysentechnik), then deflected by two galvanometric scanning mirrors (6215HM40B, CTI-Cambridge) and directed toward the objective lens (HCX PL APO 100/1.40/0.70 Oil, Leica Microsystems) by the same set of scan and tube lenses as the ones used in a commercial scanning microscope (Leica TCS SP5, Leica Microsystems). The fluorescence light was collected by the same objective lens, de-scanned, and passed through the dichroic mirrors as well as through a fluorescence band pass filter (ET Bandpass 525/50 nm, AHF analysentechnik) before being focused (focal length 60 mm, AC254-060-A-ML, Thorlabs) into a fiber pigtailed single-photon avalanche diode (PDM Series, Micro Photon Devices, Bolzano, Italy). Photon arrival times were detected at each pixel by a time-correlated-single-photon-counting-card (SPC-830, Becker & Hickl). Synchronization was obtained from the reference signal provided by the Ti: Sapphire laser. All imaging operations were automated and managed by the software Inspector (Max Planck Innovation). For both the STED and excitation light, the average power $P$ was measured at the back aperture of the objective lens. Due to losses in the objective lens, the power at the sample is actually lower by 15 and 12% at 488 and 577 nm, respectively.

HeLa and CHO cells were cultured following standard protocols. Cells were plated on multi-well chambered cover glass (Lab-Tek II, Thermo Fisher Scientific) and let grow overnight. Cells were transiently transfected with a plasmid encoding for EGFP or tubulin-EGFP using Lipofectamine 2000 (Thermo Fisher Scientific) according to the manufacturer's protocol. Measurements were performed 24 h after transfection, at room temperature and keeping cells in a live cell imaging buffer (Live Cell Imaging Solution, Thermo Fisher Scientific). For each cell, measurements were acquired in the cytoplasm at a distance ~2 μm above the coverglass. The acquisition time $t_{acq}$ for a single measurement was set to 100 s unless indicated otherwise. Excitation power was set to 15 μW for measurement of EGFP and to 7 μW for measurements of tubulin-EGFP.

For calibration, aqueous solutions of purified EGFP (BioVision, Inc., Milpitas, CA) were prepared by diluting the protein in PBS (phosphate-buffered saline 1×, Thermo Fisher Scientific) at a final concentration of ~100 nM. For each measurement a ~10 μl drop was deposited onto multi-well chambered cover glass (the same type used for cells) previously treated for 1 h with a 1% BSA solution to prevent protein sticking to the glass. Acquisition was performed ~2 μm above the coverglass for a total acquisition time of 100 s.

An aqueous solution of EGFP with viscosity higher than water was prepared by adding glycerol to water (~15%w/w), vortexing and then diluting the protein in this mixture. An aqueous solution of a secondary antibody labeled with the dye Oregon Green 488 was prepared by diluting the secondary antibody in PBS.

**Data availability**. The data that support the findings of this study are available from the corresponding author upon reasonable request.

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

## Acknowledgements

The authors would like to thank Milka Stakic for providing the tubulin-EGFP plasmid and Enrico Gatton for useful discussions. The authors would also like to thank Roberta Pelizzoli, Arta Mehilli, and Marina Nanni for technical assistance.

## Author contributions

L.L. and G.V.: Conceived the idea. L.L., L.S., M.D.B., and G.V.: Performed experiments. F.C. and R.B.: Provided reagents. L.L. and L.S.: Wrote software. A.D. and P.B.: Assisted in the project. All authors analyzed and discussed results. L.L., F.C. and G.V.: Wrote the manuscript with input from A.D. and P.B.

## Additional information

**Competing interests:** The authors declare no competing financial interests.

