## [Peer Review File · Nature Communications]

Reviewer #1 (Remarks to the Author):

STED-FCS (i.e. the application of fluorescence-correlation-spectroscopy (FCS) on a super-resolution STED microscope) has proven to be a powerful tool for deciphering molecular diffusion and interaction dynamics with unprecedented detail (see for example Nat Commun 2014, ref 38). So far, it has solely been applied to 2D diffusion/interaction properties of molecules in membranes. Yet, studying similar properties in the cellular cytosol is equally important. Unfortunately, 3D-STED-FCS has been shown to be hampered by out-of-focus contributions, which limited cytosolic applications so far.

Lanzano et al present a clever approach to surpass this limitation and apply single-photon weighting algorithms based on the fluorescence lifetime (which the authors have applied before for the case of imaging – also published in Nature Communications) to correct for bias from out-of-focus contributions. The success of this SPLIT-STED-FLCS approach is shown for EGFP diffusion inside cells. This is a very clever and successful approach and a very important contribution to the scientific field of molecular diffusion studies – it definitely deserves publication in Nature Communications.

In parts, the manuscript is very tough to read. Some things are not well introduced or justified. For example:

- The introduction misses on the description of the diffusion law, which is an integral part of the later measurements (although law is a confusing word).
- The existing problems of 3D-STED-FCS are not straightforward described (i.e. out-of-focus undepleted signal), although it has been well described before (ref 6). This needs to be detailed in a clearer and compact way, best in the introduction – the arguments are in the manuscript already, but very scattered.
- The parameter k_s is introduced in a confusing way (page 6ff) – it did not get clear how it changes with what here. The meaning of an extraction of k_s from the data is hard to understand. On page 8, the authors state that k_s is extracted from the fit – what fit?
- Equation 2: What is J ?
- Page 9, beginning: The authors checked the removal of background here for other data – so far removal has not been tackled yet – only the determination – very confusing. The authors should stretch more the fact here that they have tested their algorithm also for another dye – this is important and deserves more focus.
- Line 164: It is not well justified why a red shift causes less background.
- Page 9: r_1 needs to be defined/introduced better.
- Lines 175ff: Not clear what three curves are meant here – also the data should be shown. Can any information be potentially extracted from the so far un-used two curves?

- Page 11: It needs to be better highlighted that uncorrelated out-of-focus background leads to a damped FCS amplitude and thus to an overestimated N – see comment above.
- The authors should plot normalized values of N/N_{conf} and τ/τ_{conf} in all cases – for free diffusion, all curves should overlap.
- Line 257: I cannot see a plot of the dependency of the diffusion coefficient on area anywhere – the authors should bring those.
- One critical issue is the very long measurement time. Is there any chance to reduce this, maybe with the limitation that not 80nm but only 100nm can be reached? The authors should highlight this, maybe with data?
- Lines 289ff: The authors should show photobleaching data also for the pure confocal case (i.e. without the STED laser) – just to exemplify how much danger the STED laser really brings along – would be interesting.
- Lines 353/354: The signal-to-noise is already low, combining the acquisition with scanning will lower it further – is this really a realistic experiment?

Minor issues:

- Introduction: The first paper to introduce STED-FCS was reference 16.
- Fitting of FCS data: It is known that EGFP shows complex photophysics, which manifests itself as additional decays in the 1-100 microsecond range of the FCS curves – I am surprised, it is not considered here.
- Line 146: What modulation?
- Line 164: What is meant by STED decay?
- Line 179: What is meant by width here?
- Line 200: $2e$ instead of $2f$.
- Line 223: Which correction factor?
- Lines 227/228: Did not get the point here?
- Line 229: Which number?
- Line 232: Not really true – reference 6 used almost isotropic volumes – still got uncorrelated background.
- Lines 235/236: The word calculate sounds very unlucky here – seems the authors are putting their number to match a “good” mobility value.
- The authors should mention the STED laser powers within the text.

- Line 244: Which calibration?
- Line 369: Should k_s not as well vary with environment – it is dependent on τ , as mentioned somewhere?
- Lines 382ff: What is T ?
- Lines 400ff: What is Δt ?
- Lines 408ff: No anomaly factor is used (as in many STED-FCS experiments before) – why?
- Line 495: What did L.L. do – pretty lonely there.
- Figure 2b: What are the different symbols?
- Supplementary Figures 1e,f: Are these curves normalized onto each other?
- Supplementary Figures 2 a and b seem to be exchanged – impossible that the 20mW data is more noisy than the 50mW one. Also the decays seem longer in b.
- Supplementary Figure 3c: What are the large volumes at the top and bottom?

Reviewer #2 (Remarks to the Author):

This manuscript by Lanzano et al. describes the application of STED-FCS to measure diffusion of EGFP in three dimensions (3D) in solution and in living cells. The authors use a combination of separation of photons by lifetime tuning (SPLIT) and fluorescence lifetime correlation spectroscopy to filter photon data by lifetime decay and code trajectories for location in the STED focal volume which effectively remove contribution from uncorrelated background. This spatio-temporal filtering using time resolved TCSPC measurements separates this approach from other applications of STED FCS allowing measurements in 3D compared to previous studies on 2D membrane systems. The authors apply this method to calibration/verification experiments of EGFP diffusing in solutions where they are able to measure diffusion coefficients and focal volume corrected fluorescent protein copy numbers. The authors also apply the method to measure the diffusion coefficients and fluorescent protein occupation numbers at a variety of spatial scales in HeLa cells and CHO cells and recover different linear diffusion laws for the two cell types.

Overall I found this work to be an elegant technical extension of STED FCS that really shows how time resolved lifetime measurements can be used to filter the photon trajectories and efficiently remove out of focus background and perform measurements in 3D. This is certainly the key novel aspect of this work.

However, I have a major concern about this paper. I feel the applications are weaker as the paper only demonstrates the measurement of EGFP diffusion and concentrations in vitro and in 3D in two tissue culture cell types (previously measured by other methods all be it not at the resolutions afforded by STED), but does not exploit the stated promise of the technique to really reveal details on confinement relevant to the cell biology or experimental treatment. While I am excited by the technical advance, I really feel strongly that the demonstration of the method to unravel biologically relevant confinement or a confinement dependent mechanism is missing and is essential.

Additionally I have a number of technical points that should addressed:

1. It would be useful to see residuals for the autocorrelation function fits to judge the quality of the fit decay models for the different scales.
2. The authors measure τ_0 independently for each cell measurement, however, they use the k -s parameter measured from the in vitro solution samples for input in the cell measurements. Can this be justified experimentally? Is it not possible that the k -s parameter which is a relative variation in the lifetime decay within the observation volume will be different in buffer solution as compared to cells (which themselves may show spatial heterogeneity for this parameter)? I feel some justification of this in vitro calibration for application in the cell measurements is necessary.
3. For the corrected number density measurements, the authors indicate that the fits are linear with zero intercepts. It is not clear whether these fits are fixed to pass through zero intercept or whether the linear fits are unconstrained and the best fits all pass through zero within error (i.e. are there offsets present in an unconstrained fit?).
4. How reproducible are diffusion measurements made from the same spot in living cells?

Minor points

Supp. Fig. 1 mW (not mw)

Supp. Fig. 1 Label or state STED versus Confocal ACF colors in the figure caption.

Supp. Fig. 4 What are the dashed lines?

Reviewer #1 (remarks to the author):

STED-FCS (i.e. the application of fluorescence-correlation-spectroscopy (FCS) on a super-resolution STED microscope) has proven to be a powerful tool for deciphering molecular diffusion and interaction dynamics with unprecedented detail (see for example Nat Commun 2014, ref 38). So far, it has solely been applied to 2D diffusion/interaction properties of molecules in membranes. Yet, studying similar properties in the cellular cytosol is equally important. Unfortunately, 3D-STED-FCS has been shown to be hampered by out-of-focus contributions, which limited cytosolic applications so far.

Lanzano et al present a clever approach to surpass this limitation and apply single-photon weighting algorithms based on the fluorescence lifetime (which the authors have applied before for the case of imaging – also published in Nature Communications) to correct for bias from out-of-focus contributions. The success of this SPLIT-STED-FLCS approach is shown for EGFP diffusion inside cells. This is a very clever and successful approach and a very important contribution to the scientific field of molecular diffusion studies – it definitely deserves publication in Nature Communications.

We thank the reviewer for all the useful comments. We modified the manuscript following the reviewer's suggestions. Below we report our point-by-point reply.

In parts, the manuscript is very tough to read. Some things are not well introduced or justified. For example:

- The introduction misses on the description of the diffusion law, which is an integral part of the later measurements (although law is a confusing word).

We agree with the referee.

We added a sentence in the introduction: “The plot of the transit time t_D of a molecule measured by FCS as a function of the observation area (the so-called FCS diffusion law), has been shown to be fundamental for discriminating between different types of motion, as for instance free diffusion versus diffusion confined by microdomains¹³.”

- The existing problems of 3D-STED-FCS are not straightforward described (i.e. out-of-focus undepleted signal), although it has been well described before (ref 6). This needs to be detailed in a clearer and compact way, best in the introduction – the arguments are in the manuscript already, but very scattered.

We agree with the referee. We modified the relative text as following:

We added a clear statement in the introduction: “The main problem encountered when performing STED-FCS in 3D is a significant increase in unspecific background signal that damps the correlation amplitude and precludes accurate FCS measurements^{6, 8}.”

We moved in the same section the arguments related to removal of uncorrelated background (e.g. “A distinctive feature of the SPLIT method, with respect to other time-resolved STED methods^{9, 19}, is the isolation of uncorrelated background¹⁸” and “In CW-STED-FCS the two most common sources of temporally uncorrelated background are (i) the detector afterpulse and (ii) the anti-Stokes fluorescence emission induced directly by the STED beam²⁰ ...”)

We added a reference to a very recent paper: (“An alternative strategy is trying to determine this background directly and subtracting it from the total signal [Gao et al, Nature Photonics 2017]”).

- The parameter k_S is introduced in a confusing way (page 6ff) – it did not get clear how it changes with what here. The meaning of an extraction of k_S from the data is hard to understand.

We apologize for that. Even though a detailed explanation of this factor was present in the Ref. [Lanzano et al, Nat Commun 2015], we agree that it needs to be explained better in the manuscript. We added the following description:

“The parameter $k_S = I_{\text{STED}}(w_0)/I_{\text{SAT}}$ is the ratio between the value of STED intensity at radial position $r = w_0$ and the saturation value I_{SAT} for which the probability of decay due to stimulated and spontaneous emission are equal²³. The precise value of k_S depends on the optical configuration, i.e. the intensity distributions, and on the properties of the sample¹⁹. In order to estimate the value of k_S for a given optical configuration and for a given sample, we introduced in Ref.23 an analytical model for the fluorescence intensity decay corresponding to the lifetime distribution described by equation (1).

On page 8, the authors state that k_S is extracted from the fit – what fit?

We added: “The value of k_S for a given STED power and for a given fluorophore can be extracted by measuring the experimental STED decay and by fitting it to equation (2).”

- Equation 2: What is J?

We apologize for that. We modified the text to be more specific: “According to this model the decay of the fluorescence intensity $J_{\text{STED}}(t)$ in presence of the STED beam (hereafter referred to as STED decay) can be described by the following formula (see Methods):”

- Page 9, beginning: The authors checked the removal of background here for other data – so far removal has not been tackled yet – only the determination – very confusing. The authors should stretch more the fact here that they have tested their algorithm also for another dye – this is important and deserves more focus.

We agree with the referee. We modified the sentence and moved it after the description of the data on Fig.1: “In order to check that the removal of background works efficiently in more critical experiments, namely when the total amount of uncorrelated photon counts is comparable with the total amount of photon counts from the decaying fluorescence signal, we also performed tests using another fluorophore (Supplementary Fig.3).”

- Line 164: It is not well justified why a red shift causes less background.

We modified the sentence and added a reference: “Since at the used STED beam wavelength (577 nm) direct excitation of EGFP is negligible [Rankin, Biophys J 2011], the uncorrelated background in these experiments is relatively low ($b < 0.03$) also for increasing STED beam power.”

- Page 9: r_1 needs to be defined/introduced better.

We agree on that. We added the following text in the description of the SPLIT-FLCS method: “The requirement of FLCS is that a decay pattern or temporal fingerprint is specified for each component. Here, in order to specify the decay patterns, we set an arbitrary value of the parameter r_1 (expressed in unit of w_0) and calculate the decay patterns associated to the inner ($r < r_1$) and outer ($r > r_1$) spatial components, according to the lifetime distribution described by equation (1) (see Methods). By decreasing the value of the parameter r_1 we are able to probe FCS on smaller observation volumes.”

- Lines 175ff: Not clear what three curves are meant here – also the data should be shown. Can any information be potentially extracted from the so far un-used two curves?

We apologize for that. We added now Suppl.Fig.1 where we show examples of the 3 ACFs obtained by FLCS analysis. We show two examples from the same data described on Fig.1c (EGFP in PBS,

STED=50mW, $r_1=11w_0$ and $r_1=0.5w_0$). The unused ACFs (components “ $r>r_1$ ” and “b”) are very noisy and, at least in our opinion, they don’t seem to contain extra information. It should be noted that, compared to SPLIT imaging (where we could clearly see a ‘ring’ PSF in the component $r>r_1$), here we probably don’t see this effect due to the lower brightness associated to component $r>r_1$.

- Page 11: It needs to be better highlighted that uncorrelated out-of-focus background leads to a damped FCS amplitude and thus to an overestimated N – see comment above.

We agree with the referee. We reorganized part of the text accordingly.

In particular we added this sentence in the introduction:

“The main problem encountered when performing STED-FCS in 3D is a significant increase in unspecific background signal that damps the correlation amplitude and precludes accurate FCS measurements^{6, 8}”

In the Results section, we added the text:

“Next we checked if the decrease of the lateral size w of the effective observation volume corresponded to a decrease in the number of molecules N calculated from the amplitude $G(0)$ of the ACFs (see Methods). A non-linear scaling emerges by plotting the number of molecules N as a function of the lateral size w^2 (Supplementary Fig.4a). In order to correct the number of molecules N , we introduce now a geometrical correction factor based on the following considerations.”

- The authors should plot normalized values of N/N_{conf} and τ/τ_{conf} in all cases – for free diffusion, all curves should overlap.

- Line 257: I cannot see a plot of the dependency of the diffusion coefficient on area anywhere – the authors should bring those.

The results have been obtained by performing only single-point STED-FCS measurement. We did not acquire a confocal FCS measurement in correspondence of each STED measurement.

However, we agree with the referee that visualization of the diffusion coefficient vs area can be helpful. We added these plots on Fig.3.

- One critical issue is the very long measurement time. Is there any chance to reduce this, maybe with the limitation that not 80nm but only 100nm can be reached? The authors should highlight this, maybe with data?

We thank the referee for raising this very interesting point. The measurement time can be reduced at the expense of the sub-diffraction spatial resolution that can be reached.

This is shown now in the revised version by new data obtained with a shorter acquisition time ($t=20s$) and analyzed to highlight the corresponding increase in relative error (new Fig.2e).

We added the following sentence in the main text:

“Finally, another factor to take into account is the acquisition time: for instance, for the measurement at a STED power of 50 mW, reducing the acquisition time t_{acq} from 100 s to 20 s causes a reduction in signal-to-noise ratio that will limit the smallest value that we can get to $w\sim 105nm$ (Fig.2e).”

We added also short acquisition data on EGFP in cells (fig.3f and g and description in the main text), to show reproducibility of the single-point diffusion law.

- Lines 289ff: The authors should show photobleaching data also for the pure confocal case (i.e. without the STED laser) – just to exemplify how much danger the STED laser really brings along – would be interesting.

We show in Suppl.Fig.9 photobleaching also for some pure confocal acquisition. We can see that photobleaching is still present even if less severe. In general we observe that photobleaching depends on both excitation and STED intensities. For this reason we modified a sentence into:

“Increasing the STED or excitation beam power is limited by the potential photo-damage induced on the fluorophores on the cells.”

- Lines 353/354: The signal-to-noise is already low, combining the acquisition with scanning will lower it further – is this really a realistic experiment?

We agree that the reduction in signal-to-noise makes combination with scanning challenging. However, we will attempt to compensate for this reduction. (e.g. by limiting the smallest spatial scale that can be reached and/or by averaging results from groups of pixels).

Minor issues:

- Introduction: The first paper to introduce STED-FCS was reference 16.

We fixed it.

- Fitting of FCS data: It is known that EGFP shows complex photophysics, which manifests itself as additional decays in the 1-100 microsecond range of the FCS curves – I am surprised, it is not considered here.

At the level of excitation of our experiments we don't observe these effects.

- Line 146: What modulation?

We substitute the word “modulated” with the word “decaying” and rewrite the whole sentence in a clearer way.

- Line 164: What is meant by STED decay?

We defined it better: “the decay of the fluorescence intensity $J_{STED}(t)$ in presence of the STED beam (hereafter referred to as STED decay)”

- Line 179: What is meant by width here?

We substituted “are characterized by smaller width” with “decay faster”

- Line 200: 2e instead of 2f.

We fixed it

- Line 223: Which correction factor?

the correction factor s defined by equation (3). We specified it now

- Lines 227/228: Did not get the point here?

We modified the sentence into: “The experimental values of s are in perfect agreement with equation (22), that was derived within this model (Fig.2f).”

- Line 229: Which number?

Particle number; we specified it now

- Line 232: Not really true – reference 6 used almost isotropic volumes – still got uncorrelated background.

We agree on that. We modified the sentence into: “It would be interesting to check if a similar correction works also on STED configurations able to generate a more isotropic lifetime pattern in 3D^{8,25},”

- Lines 235/236: The word calculate sounds very unlucky here – seems the authors are putting their number to match a “good” mobility value.

We agree. We substituted it with the word ‘measure’

- The authors should mention the STED laser powers within the text.

We fixed it

- Line 244: Which calibration?

Calibration with EGFP in PBS solution; we specified it now

- Line 369: Should k_s not as well vary with environment – it is dependent on τ , as mentioned somewhere?

We assume that the STED saturation intensity of the fluorophore I_{SAT} is the same in solution and in cells. We stated it more clearly in the revised version.

- Lines 382ff: What is T ?

We already defined T on page 6 as “the period of excitation T ($T=12.5$ ns)”. We defined it again for better clarity.

- Lines 400ff: What is Δt ?

Δt was defined on line 393 as the bin width

- Lines 408ff: No anomaly factor is used (as in many STED-FCS experiments before) – why?

We didn't use the anomaly factor because data could be fitted to a simpler model.

- Line 495: What did L.L. do – pretty lonely there.

Thanks, we fixed it.

- Figure 2b: What are the different symbols?

The values of k_s and b recovered from the curves shown in part (a). We added two arrows on the plot.

- Supplementary Figures 1e,f: Are these curves normalized onto each other?

Yes, we normalized the curves for a better visualization. We corrected the caption.

- Supplementary Figures 2 a and b seem to be exchanged – impossible that the 20mW data is more noisy than the 50mW one. Also the decays seem longer in b.

This should not be surprising by looking at the size of the effective volumes. We can reach 100nm with 80 mW (panel b, 3rd ACF from top to bottom) with very good signal-to-noise. We can reach 100nm also with only 20mW (panel a, last ACF from top to bottom) but with a much higher level of noise.

- Supplementary Figure 3c: What are the large volumes at the top and bottom?

The large volumes correspond, schematically, to the out-of-focus regions of longer lifetime. The SPLIT-FLCS algorithm operates by selecting the photons with longer lifetime and this results in an asymmetric SPLIT effective volume.

Reviewer #2 (Remarks to the Author):

This manuscript by Lanzano et al. describes the application of STED-FCS to measure diffusion of EGFP in three dimensions (3D) in solution and in living cells. The authors use a combination of separation of photons by lifetime tuning (SPLIT) and fluorescence lifetime correlation spectroscopy to filter photon data by lifetime decay and code trajectories for location in the STED focal volume which effectively remove contribution from uncorrelated background. This spatio-temporal filtering using time resolved TCSPC measurements separates this approach from other applications of STED FCS allowing measurements in 3D compared to previous studies on 2D membrane systems. The authors apply this method to calibration/verification experiments of EGFP diffusing in solutions where they are able to measure diffusion coefficients and focal volume corrected fluorescent protein copy numbers. The authors also apply the method to measure the diffusion coefficients and fluorescent protein occupation numbers at a variety of spatial scales in HeLa cells and CHO cells and recover different linear diffusion laws for the two cell types.

Overall I found this work to be an elegant technical extension of STED FCS that really shows how time resolved lifetime measurements can be used to filter the photon trajectories and efficiently remove out of focus background and perform measurements in 3D. This is certainly the key novel aspect of this work.

However, I have a major concern about this paper. I feel the applications are weaker as the paper only demonstrates the measurement of EGFP diffusion and concentrations in vitro and in 3D in two tissue culture cell types (previously measured by other methods all be it not at the resolutions afforded by STED), but does not exploit the stated promise of the technique to really reveal details on confinement relevant to the cell biology or experimental treatment. While I am excited by the technical advance, I really feel strongly that the demonstration of the method to unravel biologically relevant confinement or a confinement dependent mechanism is missing and is essential.

We thank the reviewer for all the useful comments. In response to the reviewer's major concern, we now revised the manuscript in such a way that the potential of the method to reveal details on confinement could be better highlighted.

In this respect, we would like to draw the attention of the reviewer to the following points:

1) Already in the original version, looking at the behavior of EGFP in the cytoplasm, one could see that some points showed confinement (old Fig3b), as revealed by the non-zero intercept, even if the average 'diffusion law' (old Fig3d) indicated a non-confined diffusion. We think that this observation is important, and highlights the unique capability of the method to detect spatial heterogeneity in the diffusion mode of EGFP in the cytoplasm.

In order to stress this point, we modified Fig.3 by rearranging the order of the plots and by adding new data representing diffusion measurement performed consecutively on the same point (as suggested by the referee in another comment).

2) Following the suggestion of the referee, we investigated the behavior of another protein (tubulin-EGFP) which, compared to EGFP, is expected to be, at least transiently, bound/trapped to the locations where microtubules are present. The new data are reported in Figure 4. In this case we show that

trapping is already visible in the average diffusion law (Fig.4b), which shows, in contrast to EGFP, a positive intercept.

Even more interesting are the results from the single-point measurements that are now much more scattered compared to the case of EGFP (Fig.4c). By looking at the single-point 'diffusion laws' we can distinguish points where trapping of the tubulin molecule is dominating and points where trapping is negligible.

To the best of our knowledge, this is the first time that trapping of a molecule (tubulin in this case) is observed on an arbitrary point in the cytoplasm of live cells by STED-FCS.

Additionally I have a number of technical points that should be addressed:

We also thank the reviewer for these additional comments. Our reply is reported below.

1. It would be useful to see residuals for the autocorrelation function fits to judge the quality of the fit decay models for the different scales.

We added the residuals using the right axis of the same plots.

2. The authors measure τ_0 independently for each cell measurement, however, they use the k_s parameter measured from the in vitro solution samples for input in the cell measurements. Can this be justified experimentally? Is it not possible that the k_s parameter which is a relative variation in the lifetime decay within the observation volume will be different in buffer solution as compared to cells (which themselves may show spatial heterogeneity for this parameter)? I feel some justification of this in vitro calibration for application in the cell measurements is necessary.

If we look at previous works on STED-FCS, the size of the effective volume, at a given STED power, is calibrated using a sample containing the free-diffusing fluorophore or by imaging fluorescence beads with a similar emission spectrum. In other words, it is implicitly assumed that the level of depletion achieved on the calibration sample is the same as the one in the actual sample (membrane, cell, etc).

In our case, the parameter $k_s = I_{\text{STED}}(w_0) / I_{\text{SAT}}$, is the ratio between the value of STED intensity at radial position $r = w_0$ and the saturation value I_{SAT} for which the probability of decay due to stimulated and spontaneous emission are equal. The value $I_{\text{STED}}(w_0)$ depends on the used STED power and the optical configuration (i.e. donut PSF). The value I_{SAT} is the saturation intensity value that depends on the fluorophore and the STED wavelength.

In line with previous works on STED-FCS, we assume that the value I_{SAT} (and thus the value of k_s) is approximately the same for EGFP in solution and in cells.

For the sake of clarity, in the revised manuscript, we introduced better the parameter k_s (in the Results section) and explained this approximation in the Methods section.

3. For the corrected number density measurements, the authors indicate that the fits are linear with zero intercepts. It is not clear whether these fits are fixed to pass through zero intercept or whether the linear fits are unconstrained and the best fits all pass through zero within error (i.e. are there offsets present in an unconstrained fit?).

We apologize for that. In the revised version, the fits in Fig3 and Fig4 are unconstrained linear fits, so that the scaling of the number in the two cases can be compared.

4. How reproducible are diffusion measurements made from the same spot in living cells?

We thank the referee for this comment. We added in Fig.3f,g new data representing three shorter acquisition ($t=20s$) performed consecutively on the same point. Please note that, by reducing the acquisition time from 100s to 20s, we cannot reach the same spatial scale (this is now explained in the new Fig.2 and in the text). However, the data clearly show that the anomaly of the diffusion in that location (negative intercept) is persistent in time. This confirms that the variability in the single-point 'diffusion laws' shown in Fig3d is actually due to spatial heterogeneity of the cytoplasm.

Minor points

Supp. Fig. 1 mW (not mw)

We fixed it

Supp. Fig. 1 Label or state STED versus Confocal ACF colors in the figure caption.

We fixed it

Supp. Fig. 4 What are the dashed lines?

We added a sentence in the caption

Reviewer #1 (Remarks to the Author):

The authors have very well commented on all of my concerns (and to my opinion also to the other referee's concerns) and revised the manuscript accordingly. It reads very well now. This is a very important study with potential high impact – I strongly support publication in Nature Communications in its current form.

I have a few very minor comments that the authors might want to consider prior to publication.

- Line 52: Maybe write "...transit time t_D of a molecule through the observation volume ..."?
- Line 64: One could cite reference 18 already here.
- Line 93: Maybe write "... to perform background-unbiased STED-FCS ..."?
- Line 102: Maybe add "... FCS-FIDA analysis or separate background determination"?
- Figure 4d: It is confusing that $N\#$ does not originate at 0 – it should from theory, since it is not influenced by the diffusion mode. Maybe the authors want to leave this plot out – otherwise they have to come up with an explanation?
- Lines 350ff: The authors mix up past and present here – they should stay consistent with the previous parts and describe the experiments in past mode.
- Figure 1a: Label x-axis.
- Figure 2: Name dashed red line in c, and blue, grey dashed and red dashed lines in d in captions.
- Figure 3b: Name open and solid squares in caption.

Reviewer #2 (Remarks to the Author):

I feel the revised version of this manuscript is clearer and the additional tubulin measurements add more novelty to the demonstration of this method. I am in favor of publication of the revised version.

Very minor points

line 60 pg 6 and also line 325 page 17 the authors refer to bidimensional regarding the membrane. It is more typical to refer to two dimensional which is also consistent with the later reference to three dimensional environments.

linge 149 pg 9 in unit[s] of ω_0 (add s to units)

Reviewer #1 (remarks to the author):

The authors have very well commented on all of my concerns (and to my opinion also to the other referee's concerns) and revised the manuscript accordingly. It reads very well now. This is a very important study with potential high impact – I strongly support publication in Nature Communications in its current form.

We thank the reviewer for the useful comments. We have modified the manuscript following his/her suggestions.

I have a few very minor comments that the authors might want to consider prior to publication.

- Line 52: Maybe write "...transit time t_D of a molecule through the observation volume ..."?

We modified the text

- Line 64: One could cite reference 18 already here.

We fixed it

- Line 93: Maybe write "... to perform background-unbiased STED-FCS ..."?

We modified the text

- Line 102: Maybe add "... FCS-FIDA analysis or separate background determination"?

We modified the text

- Figure 4d: It is confusing that $N\#$ does not originate at 0 – it should from theory, since it is not influenced by the diffusion mode. Maybe the authors want to leave this plot out – otherwise they have to come up with an explanation?

We moved the plot to Supplementary Fig.9

- Lines 350ff: The authors mix up past and present here – they should stay consistent with the previous parts and describe the experiments in past mode.

We fixed it

- Figure 1a: Label x-axis.

We fixed it

- Figure 2: Name dashed red line in c, and blue, grey dashed and red dashed lines in d in captions.

We fixed it

- Figure 3b: Name open and solid squares in caption.

We fixed it

Reviewer #2 (Remarks to the Author):

I feel the revised version of this manuscript is clearer and the additional tubulin measurements add more novelty to the demonstration of this method. I am in favor of publication of the revised version. We thank the reviewer for the useful comments. We have modified the manuscript following his/her suggestions.

Very minor points

line 60 pg 6 and also line 325 page 17 the authors refer to bidimensional regarding the membrane. It is more typical to refer to two dimensional which is also consistent with the later reference to three dimensional environments.

We fixed it

line 149 pg 9 in unit[s] of ω (add s to unit

We fixed it